# SM2-Based Offline/Online Efficient Data Integrity Verification Scheme for Multiple Application Scenarios

**DOI:** 10.3390/s23094307

**Published:** 2023-04-26

**Authors:** Xiuguang Li, Zhengge Yi, Ruifeng Li, Xu-An Wang, Hui Li, Xiaoyuan Yang

**Affiliations:** 1State Key Laboratory of Integrated Service Networks, Xidian University, Xi’an 710126, China; lixiuguang00@126.com; 2Cryptographic Engineering College, Chinese People’s Armed Police Force Engineering University, Xi’an 710086, China; rfli46@163.com (R.L.); wangxazjd@163.com (X.-A.W.); yxyangyxyang@163.com (X.Y.); 3Key Lab of the Armed Police Force for Network and Information Security, Xi’an 710086, China; yi257172@163.com

**Keywords:** cloud storage, data integrity, public auditing, SM2

## Abstract

With the rapid development of cloud storage and cloud computing technology, users tend to store data in the cloud for more convenient services. In order to ensure the integrity of cloud data, scholars have proposed cloud data integrity verification schemes to protect users’ data security. The storage environment of the Internet of Things, in terms of big data and medical big data, demonstrates a stronger demand for data integrity verification schemes, but at the same time, the comprehensive function of data integrity verification schemes is required to be higher. Existing data integrity verification schemes are mostly applied in the cloud storage environment but cannot successfully be applied to the environment of the Internet of Things in the context of big data storage and medical big data storage. To solve this problem when combined with the characteristics and requirements of Internet of Things data storage and medical data storage, we designed an SM2-based offline/online efficient data integrity verification scheme. The resulting scheme uses the SM4 block cryptography algorithm to protect the privacy of the data content and uses a dynamic hash table to realize the dynamic updating of data. Based on the SM2 signature algorithm, the scheme can also realize offline tag generation and batch audits, reducing the computational burden of users. In security proof and efficiency analysis, the scheme has proven to be safe and efficient and can be used in a variety of application scenarios.

## 1. Introduction

Cloud storage technology is convenient and flexible, its use growing rapidly at home and abroad [1]. Big data from Internet of Things (IoT) devices and medical big data also use cloud storage technology to provide services. However, after users have stored data in the cloud, although they can thereby access convenient storage and management services, they also lose the power to control the data directly. Therefore, ensuring data integrity in the cloud has become a hot research topic for scholars [2]. Data integrity verification technology uses cryptography-related technology to design appropriate schemes that convince users that their data, when stored in the cloud server, is secure and complete, by means of a series of interactions between the auditor and the cloud server. Using this technique can effectively deter cloud service providers (CSP) from deliberately concealing the issues of data loss or corruption from users due to their fear of damaging their reputations. It also effectively stops users from unreasonably making accusations or claims against CSPs simply because of suspicion, thus effectively protecting the legitimate rights of both users and CSPs [3].

IoT devices have been widely used and have become a convenient and universal access terminal for Internet services. However, IoT devices have limited storage space and weak computing power to support complex data computing and big data storage [4]. Therefore, the cloud, with its powerful computing potential and storage capacity, is generally used to expand the functions of IoT devices so that IoT devices can obtain massive data storage and strong data analysis capabilities. As one of the main service areas of cloud computing, the cloud storage mode of IoT device data allows IoT users to store their data in the cloud to compensate for the lack of storage space on IoT devices [5]. However, with the loss of the physical ownership and control of outsourced data, IoT device users are concerned about the integrity of their data. Therefore, it is necessary to conduct an integrity audit on the data of IoT devices using the cloud storage mode. Compared with general cloud audit schemes, the design of data audit schemes for the IoT has higher requirements [6]. First, public verification is required. IoT devices are often resource-constrained and need to support complex calculations; therefore, audit schemes require a third-party auditor (TPA) to be able to verify data integrity on behalf of the users. Second, privacy protection is needed. Data privacy protection is the most important problem in the IoT’s cloud auditing scheme. Over the duration of the scheme’s implementation, the contents of the challenge file should remain confidential to the TPA. Many IoT-embedded devices will generate a large quantity of personal and private data information. If this sensitive information is exposed in the integrity verification process, the privacy of IoT users may be disclosed to the integrity verifier or to the public. Third, the scheme is required to be lightweight. Computing capacity, storage capacity, bandwidth, and other resources of IoT devices are often greatly limited. Thus, audit schemes with lower computing costs are more suitable for the IoT. Fourth, a batch audit is required. There are many types and numbers of IoT devices. The audit scheme must support batch audits for multiple users to quickly verify the integrity of the massive amount of IoT data.

The sources of healthcare-derived big data mainly include clinical big data generated during patients’ medical treatment, health-related big data generated by wearable human health-monitoring devices, and biological big data generated by life sciences research and medical institutions. However, despite the large amount of data stored in medical databases, it is still not easy to comprehensively record information on all diseases. Since electronic medical records are not fully available, a large amount of data comes from manual records. Biases and incomplete content arising from the recording process, uncertainty in textual expressions, and incomplete data storage are the root causes of incomplete medical and health big data, so it is crucial to audit the integrity of medical data [7]. In addition, the integrity audit scheme of medical data needs to achieve privacy protection. Detailed personal information and the health status of patients are often directly recorded in healthcare big data, and these sensitive forms of data require greater privacy protection. Finally, dynamic update capabilities are also necessary. Patients’ consultation and onset times involve process changes, while the waveform and image data of the medical examination are time series. The patient’s health status is not static but is always in a state of dynamic change.

**Motivation**. We believe that it is essential and urgent to design a data integrity verification scheme that can be better applied to the cloud storage environment of IoT data and medical data, and the most appropriate scheme must meet the following functions:(1)Public auditing: anyone can perform the audit. Generally, experienced and skilled TPAs are entrusted by the users to perform the audit task.(2)Dynamic updating of cloud data: users can insert, delete, and modify the data stored in the cloud at any time.(3)Privacy protection: the TPA cannot know the contents of the user data. It is also preferable that CSP should not know the contents of the user data.(4)Lightweight computation: the users’ computational overhead should be as small as possible.(5)Batch audits for multiple users: the most appropriate scheme is able to implement batch audits for multi-user data.

However, we found that most existing cloud storage schemes do not meet the above five conditions well. Therefore, we designed an efficient offline/online data integrity verification scheme. The proposed scheme is not only applicable to the integrity audit of cloud data but is also applicable to the integrity verification of IoT data and medical data.

## 2. Related Works

In early remote data integrity verification schemes, the auditor needs to download all data from the cloud and use the locally stored metadata to confirm the integrity, which requires high communication and calculation costs and takes a long time to achieve, resulting in a great waste of computing power. In 2007, Ateniese et al. [8] proposed the first provable data possession (PDP) scheme. Their scheme divided the data files into blocks. The auditor only needed to download partial data blocks from the CSP to verify the integrity of all data, with a high probability. For 1,000,000 4 KB blocks, assuming that 1% of the blocks have been deleted or tampered with by the CSP, the auditor only needs to verify the integrity of 460 blocks to judge the integrity of all data with a greater than 99% confidence probability. In 2007, Juels et al. [9] first proposed the proofs of a retrievability scheme to audit data. Their scheme used error correction codes and sampling detection technology to recover the damaged data after detecting that the integrity of the cloud data was damaged. However, their scheme does not support public auditing, and the number of audits is limited.

With the increasing demands of users, scholars have expanded various functions based on the scheme proposed by Ateniese et al. In their study [8], a dynamic data updating function is added to the cloud audit scheme to enable users to modify the data stored in the cloud more flexibly. If the cloud data are directly modified, the tag and index will not match, and subsequent verification work cannot be completed. Therefore, various appropriate data structures are proposed to achieve dynamic data updates. In order to prevent malicious auditors from colluding with CSPs or stealing users’ data privacy, the random mask technology and blockchain technology are combined in cloud audit schemes to achieve security goals. In order to enable auditors to audit the data integrity of more than one user at a time, the batch audit function is added to the cloud audit scheme, which improves the efficiency of large-scale audits. In meeting the needs of one user after another, cloud data audit schemes gradually become more mature. However, with cloud storage technology, the existing cloud audit schemes are no longer fully applicable to the cloud storage environment for IoT and medical data.

The cloud audit scheme proposed in [10] constructs a multi-leaf authentication method based on the Merkle tree. The scheme can simultaneously authenticate multiple leaf nodes and realize batch data updates. The proposed scheme also supports log auditing. Users can verify whether the auditors perform their audit work honestly by checking the log files generated by auditors. However, the scheme does not mention comprehensive privacy protection, and there is a security problem wherein attackers can forge data tags to pass the audit. Hou et al. [11] designed a public audit protocol supporting blockless verification and batch verification practices; the protocol uses a chameleon certification tree to implement the efficient dynamic operations of outsourcing data, reduces the computational cost caused by data updates, and further improves audit efficiency. Nevertheless, the scheme does not describe how to achieve privacy protection for users and requires the computation of many bilinear pairs during the upload block verification and bulk audit phases. Based on the BLS signature, Mishra et al. [12] used a binomial binary tree and an indexed hash table data structure to construct an efficient and dynamically updated cloud audit scheme. However, the scheme cannot achieve batch audits.

Fan et al. [13] built a flexible auditing scheme that supports efficient dynamic updating based on the alliance blockchain. However, the scheme does not consider the batch audits of large-scale users. The ID-based offline/online PDP protocol that was constructed in [14] is based on an offline/online signature. The scheme supports batch verification and entire dynamic data operation but cannot realize data content privacy protection for cloud servers. The audit scheme introduced in [15] is based on an ID with compressed cloud storage, and it only uses encrypted data blocks in a self-verified way to audit the cloud data. Xu et al. [16] introduced the concept of transparent integrity auditing. They proposed a concrete scheme, based on the blockchain, which does not rely on third-party auditors while freeing users from high communication costs in data integrity auditing.

Ji et al. [17] proposed an ID-based data integrity verification scheme with the designated auditor. In their scheme, only the auditor designated by the user could join the audit task, which improved the scheme’s security compared with the previous ID-based audit schemes. However, the scheme needed to be more comprehensive. Li et al. [18] proposed an audit scheme based on a redactable signature. CSP can transform the signature directly, without the additional sanitizer, while sharing sensitive data. The signature can also be used to authenticate the source of sharing data. Lin et al. [19] proposed a consortium blockchain-based audit protocol. This protocol can check the abnormal behavior of auditors, but the scheme needs to be more comprehensive to achieve batch audits. In addition, during the audit process, the above schemes used numerous high-cost operations, such as the power index, point hash function, and bilinear mapping, thereby incurring high computing costs; thus, it cannot be applied to the environment of IoT data and medical data cloud storage completely.

**Our Contributions**. In this paper, we propose an efficient offline/online data integrity verification scheme for multiple application scenarios. Our contributions can be summarized as follows:(1)Based on the SM2 signature algorithm and the SM4 block encryption algorithm, we have constructed an offline/online remote data audit scheme. The scheme supports dynamic data updates, comprehensive privacy protection, and batch audit capability. Based on the advantages of offline tags and scheme design, our scheme has low computational overheads and is suitable for lightweight environments.(2)We have carried out a security analysis and proof of the scheme. The scheme is resistant to forgery attacks from the storage side and achieves comprehensive privacy protection; even the storage side cannot obtain the real content of the data.(3)We analyzed the scheme’s efficiency and compared the functions and computing costs with the existing schemes, proving the comprehensiveness of the scheme’s functions and its high efficiency.

**Organization.** We have organized the rest of this paper as follows. Section 3 introduces the system model and the security model. The background knowledge used in the scheme’s construction process and defines the proposed scheme’s system and security model are introduced in Section 4. In Section 5, the concrete scheme is described. We analyze the scheme’s performance and compared it with other schemes in Section 6. In Section 7, we conclude our work. We analyze the security of the scheme in Appendix A.

## 3. The System Model and Security Model

### 3.1. System Model

The system model of the scheme is shown in Figure 1. Three interacting entities are included: the CSP provides data storage services to users for payment, but it is not trusted and may delete data from the cloud or pry into the data privacy of its users for profit. The data owner (DO) is the owner of the data, uploading the data to the cloud to save their own storage overhead, but does not want the data privacy to be compromised. The TPA is a semi-honest auditor commissioned by users. They will faithfully perform the task of auditing the integrity of the data in the cloud, on the one hand, but on the other hand, they are curious about the content of the data.

The operation process of the proposed audit scheme includes the following algorithms:

(1)Setup: the CSP runs the algorithm, which inputs the security parameter, λ, and generates the public parameters {E,G,q,g}.(2)KeyGen: the DO runs the algorithm, which outputs the private key, $k_{s}$, and the public key, $k_{p}$.(3)OffTagGen: the DO runs the algorithm, which inputs $k_{s}$ and the random numbers $d_{i}$, l, outputting the offline tags, $r_{i}^{\prime },s_{i}^{\prime }$.(4)OnTagGen: the DO runs the algorithm, which inputs $r_{i}^{\prime },s_{i}^{\prime }$ and data blocks $m_{i}$, then outputs the online tags $r_{i},s_{i}$.(5)ChalGen: the TPA runs the algorithm, which inputs the random number π and outputs the indexes, ${\left\{i_{j}\right\}}_{\left(1\leq j\leq c\right)}$.(6)ProofGen: the CSP runs the algorithm, which inputs the ${\left\{m_{i_{j}},r_{i_{j}},s_{i_{j}},i_{j}\right\}}_{\left(1\leq j\leq c\right)}$ and outputs the proof {ρ,s,r}.(7)VerifyProof: the TPA runs the algorithm, which inputs the proof {ρ,s,r} and outputs “true” or “false” to indicate the integrity of the data.

### 3.2. Security Model

In the existing data integrity audit schemes, security analysis often considers the CSP to be unreliable; it will forge tags in an attempt to pass the audit. Therefore, we mainly prove the unforgeability of the current scheme in the security analysis; this means that if the DO’s data are corrupted, this must be detected by the interaction between the CSP and TPA when executing the scheme. That is, the CSP cannot forge integrity evidence and pass the data integrity audit under the condition that the data security is damaged; thus, it must carefully maintain the cloud data. We can define the unforgeability of the scheme with the following game:

Game: Assuming that C is the challenger, C runs the Setup algorithm to generate the system parameters and sends the system parameters to an adversary, A. In this security model, we assume that the adversary A has great privileges, although these privileges are unlikely to be possessed in a real situation. In Appendix A, we will show that even if the adversary, A, has all the privileges assumed herein, he/she is unable to break the auditing scheme proposed in this paper, thus demonstrating that the scheme has high security strength. Except for the target user that adversary A wants to attack, he/she can inquire about any other user’s information. Specifically, A can ask the following predictor:(1)Public key query: When A queries the public key of ${\mathrm{ID}}_{w}$, C runs the KeyGen algorithm to generate $k_{wp}$ and returns $k_{wp}$ to A.(2)Private key query: When A queries the public key of ${\mathrm{ID}}_{w}$, C runs the KeyGen algorithm to generate $k_{ws}$ and returns $k_{ws}$ to A.(3)Tags query: A can obtain the tag of $m_{wi}$ under the public key $k_{wp}$ of ${\mathrm{ID}}_{w}$.

Based on the above query, after A is challenged, if A outputs the aggregate tag $\left\{\rho_{w}^{*},s_{w}^{*},r_{w}^{*}\right\}$ with the ${\mathrm{ID}}_{w}^{*}$, $k_{wp}{}^{*}$, and the following conditions are met, then A wins the game. That is, our scheme is forgery-resistant.

Condition 1: The forged aggregation tags $\left\{\rho_{w}^{*},s_{w}^{*},r_{w}^{*}\right\}$ meet the verification equations.

Condition 2: There is no interruption of the public key query.

Condition 3: All the blocks $m_{wi}^{*}$ of ${\mathrm{ID}}_{w}^{*}$ have been queried tags.

## 4. Preliminaries

### 4.1. Chinese Commercial Cryptography Algorithm

In 2010, the State Cryptography Administration of China released the elliptic curve-based SM2 cryptographic algorithm. The SM2 algorithm has high cryptographic complexity, fast processing speed, lower machine performance consumption, better performance, and more security. Its security has been proven by the authors of [20], and SM2 is more secure against generalized key substitution attacks. In 2012, the Security Commercial Code Administration Office of China released the SM4 block cipher standard. This is similar to AES-128, with simplified round key generation, and it is mainly used for data encryption. The encryption algorithms and decryption algorithms both use 32 rounds of a nonlinear iterative structure, the S box is a fixed 8-bit input and 8-bit output, the number of calculation rounds is large, and nonlinear changes are added, which make them more effective in defending against key-leaking Trojans [21]. The SM2/4 algorithm has been incorporated into the ISO/IEC international standard. Given its excellent security and performance, it is believed that it will be recognized or adopted by more and more organizations and individuals in China or outside of China.

Our scheme uses the SM2 digital signature algorithm to construct the audit scheme and the specific steps of the SM2 digital signature algorithm are as follows [22]. To facilitate understanding, we define and explain the various notations that appear in this paper in Table 1.

(1)Key generation: the selected elliptic curve equation is $y^{2}=x^{3}+ax+b$. Let g be the base point on the elliptic curve; the integer $k_{s}\in Z_{q}^{*}$ is randomly selected as the private key, then the public key $k_{p}=k_{s}\cdot g$ is calculated.(2)Signature: Let the data to be signed be m. The signer first selects a random integer $d\in Z_{q}^{*}$, sets $d\cdot g=\left(x_{1},y_{1}\right)$, and computes $r=m+x_{1}$, $s={\left(1+k_{s}\right)}^{-1}\left(d-rk_{s}\right)$; the signature of the message m is {r,s}.(3)Verification: After receiving m and {r,s}, the verifier calculates t=r+s, $\left(x_{1},y_{1}\right)=sg+tk_{p}$, and $r^{\prime }=x_{1}+m$. If the values of $r^{\prime }$ and r are equal, the signature is correct.

### 4.2. Dynamic Hash Table

Our scheme uses the dynamic hash table data structure proposed in Reference [23] to achieve a dynamic update of the data in the cloud. The dynamic hash table is a two-dimensional data structure, as shown in Figure 2.

The table includes both file and data block elements. In the file element, NO. indicates the index value of the corresponding file, while ID indicates the identification of the corresponding file and a pointer of the first data block of this file. In the data block element, $t_{i}$ indicates the timestamp of the data block, and $v_{i}$ indicates the version number of the data block. The version number is initially set to 1 and its value is incremented by 1 for each change of the data block. The data block elements in the dynamic hash table are connected by a chain table, and each data block element is a node in the chain table, while each node includes the version information of the data block, the timestamp, and a pointer to the next node. Once the dynamic hash table is established, operations such as search, insert, deletion, and modification can be performed at either the file level or the data block level.

### 4.3. Elliptic Curve Discrete Logarithm Problem

The elliptic curve discrete logarithm problem (ECDLP): Let G be an additive cyclic group of elliptic curves of the order of the large prime q and set g∈G as a generator. ECDLP means that, given g,a⋅g∈G, an attacker A calculates $a\in Z_{q}^{*}$. The probability that the attacker A can solve the ECDLP in polynomial time is negligible:(1)$$\mathrm{Pr}\left[A\left(ag,g\right)=a:a\overset{R}{\leftarrow }Z_{q}^{*}\right]\leq \varepsilon$$
where ε represents the negligible probability; that is, it is computationally infeasible to solve the ECDLP.

## 5. SM2-Based Offline/Online Efficient Data Integrity Verification Scheme

In this section, we give a detailed description of the proposed scheme.

(1)Setup(λ)→(E,G,q,g): the CSP inputs the security parameter λ and generates the public parameters {E,G,q,g}. $E:y^{2}=x^{3}+ax+b\mathrm{mod}p$ is the elliptic curve, p and q are large prime numbers, G is an additive cyclic group of order q defined on E, and g is the generator of the group, G.(2)$\mathrm{KeyGen}\to (k_{s},k_{p})$: the DO randomly selects $k_{s}\in Z_{q}^{*}$ as the private key and calculates $k_{p}=k_{s}\cdot g\in G$ as the public key.(3)$\mathrm{OffTagGen}(k_{s},d_{i},l)\to (r_{i}^{\prime },s_{i}^{\prime })$: we set the number of blocks for the file to n, and the block processing can improve the calculation efficiency and realize sampling verification. The DO randomly selects ${\left\{d_{i},l\in Z_{q}^{*}\right\}}_{1\leq i\leq n}$, calculates $D_{i}=d_{i}\cdot g\in G$, and sets the coordinates of $D_{i}$ to $\left\{x_{i},y_{i}\right\}$. For i∈[1,n], the DO calculates:(2)$$r_{i}^{\prime }=x_{i}+l$$
(3)$$s_{i}^{\prime }={\left(1+k_{s}\right)}^{-1}(d_{i}-r_{i}^{\prime }k_{s})$$
and obtains the offline tag ${\left\{r_{i}^{\prime },s_{i}^{\prime }\right\}}_{1\leq i\leq n}$.(4)$\mathrm{OnTagGen}(\{r_{i}^{\prime },s_{i}^{\prime }\},m_{i})\to \{r_{i},s_{i}\}$: the DO uses the SM4 block cipher algorithm to encrypt the data file M with identity ID, and then divides M into n blocks as ${\left\{m_{i}\in Z_{q}^{*}\right\}}_{1\leq i\leq n}$, for each data block $m_{i}$. The DO generates the corresponding timestamp $t_{i}$ and version number $v_{i}$, and calculates:(4)$$r_{i}=m_{i}+r_{i}^{\prime }$$
(5)$$s_{i}=s_{i}^{\prime }-k_{s}{\left(1+k_{s}\right)}^{-1}m_{i}.$$

The DO receives the online tag ${\left\{r_{i},s_{i}\right\}}_{1\leq i\leq n}$, then sends ${\left\{\mathrm{ID},i,m_{i},r_{i},s_{i},t_{i},v_{i}\right\}}_{1\leq i\leq n}$ to CSP, sends ${\left\{\mathrm{ID},i,t_{i},v_{i},D_{i},l\right\}}_{1\leq i\leq n}$ to TPA, and finally delete the local data.

(5)$\mathrm{ChalGen}(\pi )\to \{i_{j}\}$: the TPA selects the random number $\pi \in Z_{q}^{*}$ and sends it to the cloud server. Both parties take π as input, run the same pseudo-random function, per, and obtain the random c numbers ${\left\{i_{j}\right\}}_{\left(1\leq j\leq c\right)}$ in [1,n] as the indexes of the challenged data blocks.(6)$\mathrm{ProofGen}({\left\{m_{i_{j}},r_{i_{j}},s_{i_{j}},i_{j}\right\}}_{\left(1\leq j\leq c\right)})\to \mathrm{proof}$: after the CSP receives the audit request and generates the indexes of the challenged data blocks, it calculates $\rho =\sum_{j=1}^{c}m_{i_{j}}$, $s=\sum_{j=1}^{c}s_{i_{j}}$, and $r=\sum_{j=1}^{c}r_{i_{j}}$, and sends the proof {ρ,s,r} to the TPA as the proof of data possession.(7)$\mathrm{VerifyProof}(\rho ,s,r,k_{p},D_{i_{j}},x_{i_{j}})\to \mathrm{true}/\mathrm{false}$: the TPA receives the proof {ρ,s,r}, calculates t=r+s,$D=\sum_{j=1}^{c}D_{i_{j}}$, $x=\sum_{j=1}^{c}x_{i_{j}}$, and verifies whether the following equations hold:(6)$$s\cdot g+t\cdot k_{p}=D$$(7)x+ρ+cl=r.

If Equations (6) and (7) hold, the DO is informed that the data integrity is not compromised. The correctness of them is derived as follows:(8)$$\begin{array}{l} s\cdot g+t\cdot k_{p}\\ =s\cdot g+(r+s)k_{s}\cdot g\\ =\sum_{j=1}^{c}s_{i_{j}}(1+k_{s})\cdot g+\sum_{j=1}^{c}r_{i_{j}}k_{s}\cdot g\\ =\sum_{j=1}^{c}(d_{i_{j}}\cdot g-r_{i_{j}}^{\prime }k_{s}\cdot g-k_{s}m_{i_{j}}\cdot g+(m_{i_{j}}+r_{i_{j}}^{\prime })k_{s}\cdot g)\\ =D \end{array}$$
(9)$$\begin{array}{l} x+\rho +cl\\ =\sum_{j=1}^{c}x_{i_{j}}+\sum_{j=1}^{c}m_{i_{j}}+cl\\ =\sum_{j=1}^{c}(x_{i_{j}}+m_{i_{j}}+l)\\ =\sum_{j=1}^{c}(r_{i_{j}}^{\prime }+m_{i_{j}})\\ =r \end{array}$$

(8)DynamicUpdate***:*** our scheme enables dynamic update operations on the cloud data, including insertion, deletion, and modification. Since the number of data blocks involved in the dynamic update is small, offline tags are not required in the dynamic update process. When a data block, $m_{i}$, needs to be modified to $m_{j}$, the DO selects a random number, $d_{j}$, to calculate $D_{j}=d_{j}\cdot g\in G$, where the coordinate of $D_{j}$ is set to $\left\{x_{j},y_{j}\right\}$. Then, $v_{j}$ and $t_{j}$ are generated for the data block $m_{j}$, and the tags $r_{j}=m_{j}+x_{j}+l$ and $s_{j}={\left(1+k_{s}\right)}^{-1}\cdot (k_{j}-r_{j}\cdot k_{s})$ are calculated. Finally, $\left\{\mathrm{ID},i,m_{j},r_{j},s_{j}\right\}$ and $\left\{\mathrm{ID},j,D_{j},t_{j},v_{j}\right\}$ are sent to the CSP and TPA, respectively. After receiving $\left\{\mathrm{ID},i,D_{j},t_{j},v_{j}\right\}$, the TPA finds the i-th node of the linked list corresponding to the file M in the dynamic hash table, and then replaces $v_{i}$ and $t_{i}$ with $v_{j}$ and $t_{j}$. After receiving $\left\{\mathrm{ID},i,m_{j},r_{j},s_{j}\right\}$, the CSP finds the location of $m_{i}$ and replaces $m_{i}$, $r_{i}$, $s_{i}$ with $m_{j}$, $r_{j}$, $s_{j}$.

When the DO needs to insert the data block $m_{j}$ in front of the data block $m_{i}$, they first select a random number $d_{j}$ to calculate $D_{j}=d_{j}\cdot g$ and set the coordinate of $D_{j}$ as $\left(x_{j},y_{j}\right)$. Then, they generate $v_{j}$ and $t_{j}$ for data block $m_{j}$ and calculate the tags $r_{j}=m_{j}+x_{j}+l$, $s_{j}={\left(1+k_{s}\right)}^{-1}\cdot (k_{j}-r_{j}\cdot k_{s})$. Finally, the DO sends $\left\{\mathrm{ID},i,m_{j},r_{j},s_{j}\right\}$ and $\left\{\mathrm{ID},i,D_{j},t_{j},v_{j}\right\}$ to the CSP and TPA, respectively. After receiving $\left\{\mathrm{ID},i,D_{j},t_{j},v_{j}\right\}$, the TPA finds the i−th node of the linked list corresponding to the file M in the dynamic hash table and inserts a new node after the i−th node with the content $v_{j}$, $t_{j}$. After receiving $\left\{\mathrm{ID},i,m_{j},r_{j},s_{j}\right\}$, the CSP finds the location of $m_{i}$, $r_{i}$, and $s_{i}$ according to i, ID, and inserts $m_{j}$, $r_{j}$, $s_{j}$ in front of them.

When the data block $m_{i}$ needs to be deleted, {ID,i} is sent to the CSP and TPA. After receiving {ID,i}, the TPA deletes the i−th node of the linked list corresponding to the file M in the dynamic hash table. After receiving {ID,i}, the CSP deletes $m_{i}$, $r_{i}$, and $s_{i}$ according to i.

(9)BatchAudit: the scheme can implement a batch audit for multi-user cloud data. Each DO ${\left\{u_{w}\right\}}_{1\leq w\leq x}$ randomly selects the private key, $k_{ws}\in Z_{q}^{*}$, and calculates the public key,$k_{wp}=k_{ws}\cdot g\in G$. The DO ${\left\{u_{w}\right\}}_{1\leq w\leq x}$ randomly selects ${\left\{d_{wi}^{\prime },l_{w}\in Z_{q}^{*}\right\}}_{1\leq i\leq n}$, calculates $D_{wi}=d_{wi}\cdot g\in G$, and sets the coordinates of $D_{i}$ to $\left\{x_{wi},y_{wi}\right\}$ for i∈[1,n], calculates:$r_{wi}^{\prime }=x_{wi}+l_{w}$, $s_{wi}^{\prime }={\left(1+k_{ws}\right)}^{-1}(d_{wi}-r_{wi}^{\prime }k_{ws})$, and obtains the offline tag ${\left\{r_{wi}^{\prime },s_{wi}^{\prime }\right\}}_{1\leq i\leq n}$. The DO $u_{w}$ uses the SM4 block cipher algorithm to encrypt the data file $M_{w}$ with the identity, ${\mathrm{ID}}_{w}$, and then divides $M_{w}$ into n blocks, expressed as ${\left\{m_{wi}\in Z_{q}^{*}\right\}}_{1\leq i\leq n}$; for each data block $m_{wi}$, the DO $u_{w}$ generates the corresponding timestamp $t_{wi}$ and version number $v_{wi}$, and calculates: $r_{wi}=m_{wi}+r_{wi}^{\prime }$, $s_{wi}=s_{wi}^{\prime }-k_{ws}{\left(1+k_{ws}\right)}^{-1}m_{wi}$, as the online tag ${\left\{r_{wi},s_{wi}\right\}}_{1\leq i\leq n}$, then sends ${\left\{{\mathrm{ID}}_{w},i_{w},m_{wi},r_{wi},s_{wi},v_{wi},t_{wi}\right\}}_{1\leq i\leq n}$ to the CSP, sends ${\left\{{\mathrm{ID}}_{w},i_{w},t_{wi},v_{wi},D_{wi},l_{w}\right\}}_{1\leq i\leq n}$ to the TPA, and finally deletes the local data. The TPA selects a random number π as the parameter of per and sends it to the CSP. Both sides run the same pseudo-random function, per, and obtain the random number $i_{j}^{w}{}_{\left(1\leq j\leq c\right)}$ as the index of the challenged data block. After the CSP generates the indexes of the challenged data blocks, it calculates $\rho =\sum_{w=1}^{x}\sum_{j=1}^{c}m_{wi_{j}}$,$s=\sum_{w=1}^{x}\sum_{j=1}^{c}s_{wi_{j}}$, and $r=\sum_{w=1}^{x}\sum_{j=1}^{c}r_{wi_{j}}$, then {ρ,s,r} will be sent to the TPA as the proof. The TPA receives the proof, computes t=r+s, $D=\sum_{w=1}^{x}\sum_{j=1}^{c}D_{wi_{j}}$, and $x=\sum_{w=1}^{x}\sum_{j=1}^{c}x_{wi_{j}}$, and verifies the following equations:(10)$$sg+\sum_{w=1}^{x}tk_{wp}^{}=D$$
(11)$$x+\rho +\sum_{w=1}^{x}l_{w}=r.$$

If Equations (10) and (11) hold, the TPA informs the total *x* DOs that data integrity has not been compromised. The correctness of them is derived as follows:
(12)$$\begin{array}{l} sg+\sum_{w=1}^{x}tk_{wp}\\ =\sum_{w=1}^{x}\sum_{j=1}^{c}s_{wi_{j}}g+\sum_{w=1}^{x}(k_{ws}\sum_{j=1}^{c}r_{wi_{j}}g+s_{wi_{j}}g)\\ =\sum_{w=1}^{x}((\sum_{j=1}^{c}s_{wi_{j}}g+k_{ws}s_{wi_{j}}g)+\sum_{j=1}^{c}r_{wi_{j}}k_{ws}g)\\ =\sum_{w=1}^{x}(\sum_{j=1}^{c}(1+k_{ws})s_{wi_{j}}g+\sum_{j=1}^{c}r_{wi_{j}}k_{ws}g)\\ =\sum_{w=1}^{x}\left(\sum_{j=1}^{c}(d_{wi_{j}}g-r_{wi_{j}}^{\prime }\cdot k_{ws}g-k_{ws}m_{wi_{j}}g+r_{wi_{j}}k_{ws}g\right))\\ =\sum_{w=1}^{x}\left(\sum_{j=1}^{c}(d_{wi_{j}}g-r_{wi_{j}}^{\prime }\cdot k_{ws}g-k_{ws}m_{wi_{j}}g+r_{wi_{j}}k_{ws}g\right))\\ =D \end{array}$$
(13)$$\begin{array}{l} x+\rho +c\sum_{w=1}^{x}l_{w}\\ =\sum_{w=1}^{x}\sum_{j=1}^{c}x_{wi_{j}}+\sum_{w=1}^{x}\sum_{j=1}^{c}m_{wi_{j}}+c\sum_{w=1}^{x}l_{w}\\ =\sum_{w=1}^{x}(\sum_{j=1}^{c}x_{wi_{j}}+\sum_{j=1}^{c}m_{wi_{j}}+cl_{w})\\ =\sum_{w=1}^{x}(\sum_{j=1}^{c}(x_{wi_{j}}+m_{wi_{j}}+l_{w}))\\ =r \end{array}$$

## 6. Performance Analysis

In this section, the computational overhead of the scheme and the advantage of the offline/online tags are first analyzed, then we compare the functions of our scheme with existing schemes [10,11,12,13,14], which proves that our scheme is more suitable for the IoT data storage environment and medical data storage environment. The schemes in Refs. [10,11,12,13,14] are novel cloud data audit schemes proposed in recent years. They are not out of date and, at the same time, they have been tested by scholars in the past two years. Then, we compare the computational overhead of our scheme with the schemes in Refs. [10,11,12,13,14] numerically. Finally, we experimentally verify the results of the numerical analysis of computational overhead to visualize the performance of our scheme.

We set G1 and G2 to be the additive cyclic group of $E:y^{2}=x^{3}+ax+b\mathrm{mod}p$ and the multiplicative cyclic group. p is a 512-bit prime number and q is a 160-bit prime number. The experiment was run on a 64-bit Windows 10 operating system with an i5 CPU, 2.5 GHz main frequency, and a 4 GB memory environment, using the JPBC library. After selecting a Type A elliptical curve and defining each operation, we ran each operation 10,000 times to obtain the average time overhead. The meaning of each operation and the corresponding time cost are shown in Table 2. To simplify the description, n is used here to denote the total number of data blocks, and c is used to denote the number of challenged data blocks. Because of the large values of n and c, we omit the operations’ single occurrence in our analysis of the calculation overhead.

In the OffTagGen phase, the user needs to compute $D_{i}=d_{i}\cdot g$ and $r_{i}^{\prime }=x_{i}+l$, so the computational overhead is about $n\left|M_{G1}\right|+n\left|A_{Z}\right|$. In the OnTagGen phase, the user needs to compute $r_{i}=m_{i}+r_{i}^{\prime }$ and $s_{i}=s_{i}^{\prime }-k_{s}{\left(1+k_{s}\right)}^{-1}m_{i}$, so the computational overhead is about $n\left|M_{Z}\right|+2n\left|A_{Z}\right|$. In the ProofGen phase, the CSP computes $\rho =\sum_{j=1}^{c}m_{i_{j}}$, $s=\sum_{j=1}^{c}s_{i_{j}}$, and $r=\sum_{j=1}^{c}r_{i_{j}}$, and the computational overhead is about $3c\left|A_{Z}\right|$. In the VerifyProof phase, after computing t=r+s, $D=\sum_{j=1}^{c}D_{i_{j}}$, and $x=\sum_{j=1}^{c}x_{i_{j}}$, the auditor also verifies the equations $sg+tk_{p}=D$ and x+ρ+cl=r, and the computational overhead is about $c|A_{Z}|+c|A_{G1}|$. After using the offline/online tags, the computational overhead of the user in the scheme is about $n\left|M_{G1}\right|+3n\left|A_{Z}\right|+n\left|M_{Z}\right|$. If offline/online tags are not used, the user needs to calculate $D_{i}=d_{i}\cdot g$, $r_{i}=m_{i}+x_{i}+l$ and $s_{i}={\left(1+k_{s}\right)}^{-1}\cdot (d_{i}-r_{i}\cdot k_{s})$; the computational overhead of the user is about $n\left|M_{G1}\right|+3n\left|A_{Z}\right|+2n\left|M_{Z}\right|$.

We compared our scheme with the existing certificateless schemes; the function comparison is shown in Table 3. As can be seen from Table 3, although other schemes are novel, their functions are not comprehensive. Our proposed scheme is the most comprehensive and the most suitable for the cloud storage environment of IoT data and medical data.

The numerical computational overhead comparison of our scheme and other existing schemes is shown in Table 4. In the current cloud data audit schemes, the calculation overhead of the ProofGen and VerifyProof stages is borne by the CSP and TPA, respectively, while the calculation overhead of the TagGen stage is borne by the users themselves; the users only need to bear the calculation overhead in the TagGen stage. Because of the strong computing capability of the CSP and TPA, in the design of cloud data audit schemes, more emphasis should be placed on reducing the computing cost of the user side, that is, reducing the computing cost of the audit scheme in the TagGen stage. It can be seen from Table 4 that in the TagGen stage, the computational overhead of this scheme and the scheme in [14] is the smallest and is significantly smaller than other schemes. Therefore, this scheme and the scheme in [14] are more user-friendly and can be applied to equipment with lower computational power, which is more reasonable and efficient in its design. At the ProofGen stage, the computational overhead of our scheme is also significantly lower than that of other schemes. In the case where the number of challenged data blocks, c, increases gradually, the computational overhead of the other schemes increases at a faster and more dramatic rate than that of this scheme, and the advantages of our scheme are more significant.

In order to test the performance of the scheme in terms of practical application and more intuitively compare the computational cost of each scheme, each scheme is run within the experimental environment, and the time costs in the stages of TagGen, ProofGen, and VerifyProof are recorded, as shown in Figure 3, Figure 4 and Figure 5. The number of sectors s is set at 10 [23].

Figure 3 shows the time cost of each scheme in the TagGen phase when the total number of data blocks is set to 2000, 4000, 6000, 8000, and 10,000, respectively. It can be concluded that the time cost of each scheme increases as the number of data blocks increases, but the time costs of the scheme in [14] and of our scheme do not increase significantly as the number of data blocks increases. This is due to the use of exponential operations in Refs. [10,11,12,13], which consume a significant amount of computational capacity. However, in our proposed scheme, the computation of tags is divided into two stages: OffTagGen and OnTagGen. For the users, their computation burden should mainly take into account the online tag computation. In our scheme, the online tag computation only requires simple addition and multiplication operations, resulting in a small computation overhead. Even with a large amount of data, it will not impose a significant computation burden on users. Under the conditions of the same number of data blocks, the time cost of the schemes in Refs. [10,11,12,13] is significantly higher than that of the scheme in Ref. [14] and in this scheme.

The time cost of the GenProof and VerifyProof phases is shown in Figure 4 and Figure 5, when the number of challenged blocks is set to 200, 400, 600, 800, and 1000, respectively. It can be concluded that in the GenProof stage, the time cost of the schemes in Refs. [10,14] and our scheme is relatively low, and ours is the lowest. Scheme [12] has the highest time cost. In the VerifyProof stage, the time cost of our scheme and the schemes in Refs. [10,12,14] are significantly lower than that of the schemes in Refs. [11,13]. With the increase in the number of data blocks, the audit efficiency of our scheme becomes more prominent.

According to the above performance analysis, our scheme has more comprehensive functions and less time cost at each stage, especially in the TagGen stage, so it is more compatible with lightweight devices. Therefore, our scheme is more suitable for the IoT storage environment and medical data storage environment.

## 7. Conclusions

In this paper, we constructed an efficient SM2-based offline/online data integrity verification scheme for IoT and medical data. In the stage of preprocessing data of the scheme, users use the SM4 symmetric encryption algorithm to encrypt data. We used the encrypted data to generate tags and then uploaded them to the cloud, thus achieving full data privacy protection. In the scheme, users employ the SM2 signature algorithm to construct data tags in the uploading data stage. The scheme divided tags into offline parts and online parts. Users can calculate the offline tags in advance to reduce computing costs. The scheme uses a dynamic hash table to support the dynamic update of cloud data and realizes batch audits of multi-user data. It can adapt to the IoT and medical data storage environment. The theoretical safety analysis proves the scheme’s safety. The high level of efficiency of the proposed scheme is demonstrated by comparing it with five existing schemes in terms of efficiency. In future work, we will focus on adding more functions to the existing audit schemes to meet the increasing needs of users in the cloud storage environment.

## Figures and Tables

**Figure 1 sensors-23-04307-f001:**
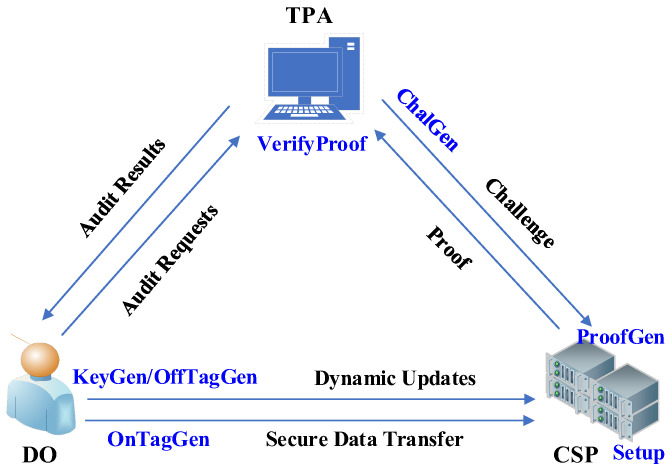
System model.

**Figure 2 sensors-23-04307-f002:**
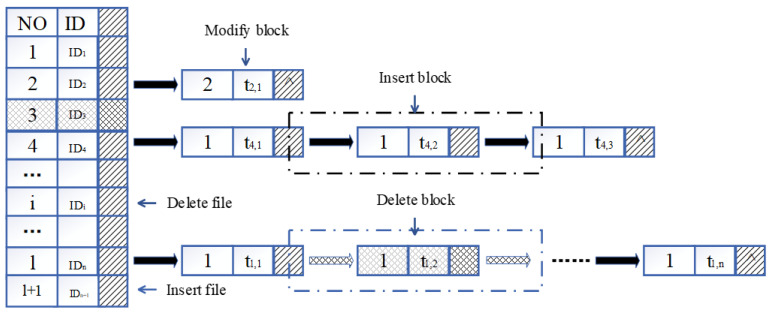
Dynamic hash table.

**Figure 3 sensors-23-04307-f003:**
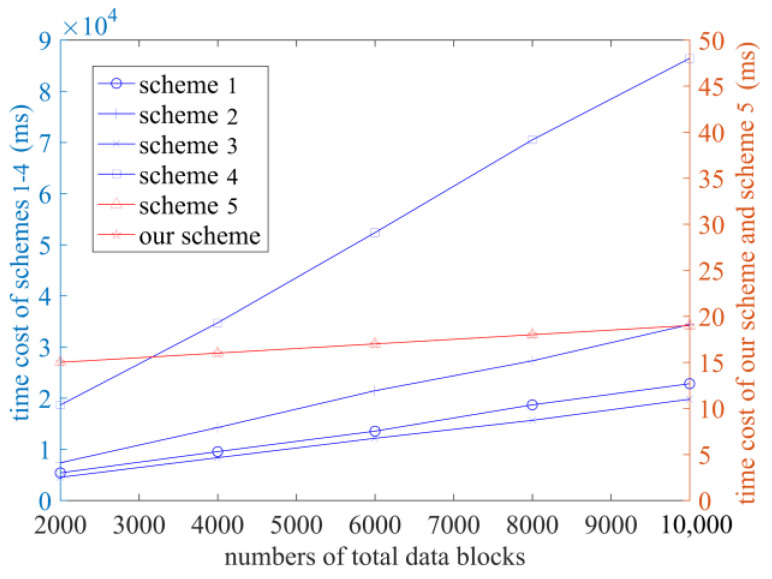
The time cost of the TagGen phase (Schemes 1–5 correspond to references [10,11,12,13,14], respectively).

**Figure 4 sensors-23-04307-f004:**
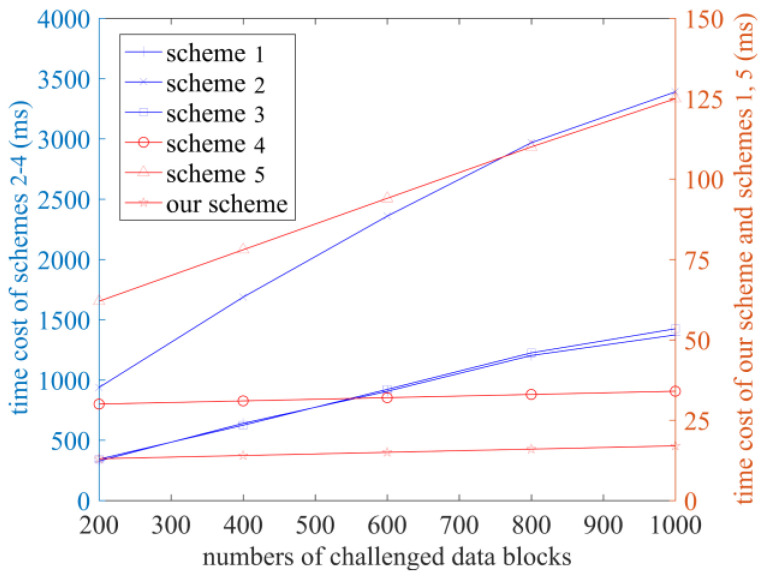
The time cost of the GenProof phase (Schemes 1–5 correspond to references [10,11,12,13,14], respectively).

**Figure 5 sensors-23-04307-f005:**
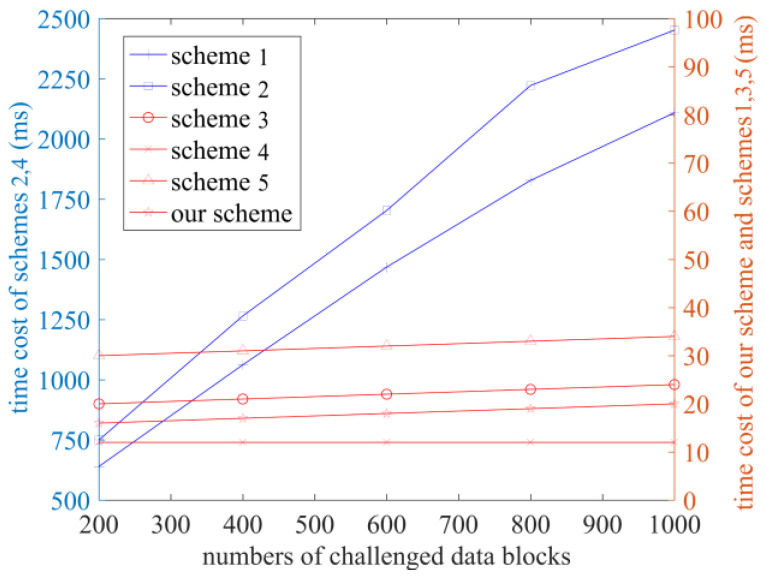
The time cost of the VerifyProof phase (Schemes 1–5 correspond to references [10,11,12,13,14], respectively).

**Table 1 sensors-23-04307-t001:** Notations used in this paper.

Notations	Descriptions
λ	The system initialization parameter.
E	The elliptic curve.
G	The additive cyclic group.
q	A large prime number.
g	G.
$k_{s}$	The user’s secret key.
$k_{p}$	The user’s public key.
$Z_{q}^{*}$	The prime field.
$d_{i},m_{i}^{\prime },l$	Random numbers.
$D_{i}$, t	Intermediate parameters.
$r_{i}^{\prime },s_{i}^{\prime }$	Offline tags.
M	The user’s data file.
$\left(m_{1}\ldots m_{n}\right)$	n data blocks.
ID	The identity of the file.
$r_{i},s_{i}$	Online tags.
$t_{i}$	$\mathrm{The}\;\mathrm{timestamps}\;\mathrm{of}\;m_{i}$.
$v_{i}$	$\mathrm{The}\;\mathrm{version}\;\mathrm{numbers}\;\mathrm{of}\;m_{i}$.
n	The number of total data blocks.
c	The number of challenged blocks.
per	The pseudo-random function.
π	The input parameter of per.
$x_{i},y_{i}$	$\mathrm{The}\;\mathrm{coordinates}\;\mathrm{of}\;D_{i}$.
ρ,s,r	The proof of data possession.

**Table 2 sensors-23-04307-t002:** Time cost of each operation.

Symbols	Description	Time Cost/ms
$\left|A_{Z}\right|$	computational cost of an addition on $Z_{q}^{*}$	0.0003
$\left|M_{Z}\right|$	computational cost of a multiplication on $Z_{q}^{*}$	0.0006
$\left|E_{Z}\right|$	computational cost of an exponentiation on $Z_{q}^{*}$	0.0226
$\left|A_{G1}\right|$	computational cost of an addition on G1	0.0055
$\left|M_{G1}\right|$	computational cost of a doubling on G1	0.7179
$\left|M_{G2}\right|$	computational cost of a multiplication on G2	0.0511
$\left|H_{Z}\right|$	computational cost of a hash operation to $Z_{q}^{*}$	0.0002
$\left|H_{G2}\right|$	computational cost of a hash operation to G2	1.1268
$\left|E_{G2}\right|$	computational cost of an exponentiation on G2	0.8107
|P|	Bilinear pair operations	5.8853

**Table 3 sensors-23-04307-t003:** Function comparison of each scheme.

	Dynamic Update	Batch Audit	Offline Tags	Privacy Protection Against the Cloud
Scheme [10]	Yes	Yes	No	No
Scheme [11]	Yes	Yes	No	No
Scheme [12]	Yes	Yes	No	Yes
Scheme [13]	Yes	No	No	Yes
Scheme [14]	Yes	Yes	Yes	No
Our scheme	Yes	Yes	Yes	Yes

**Table 4 sensors-23-04307-t004:** Comparison of the computational overhead.

	TagGen	GenProof	VerifyProof
Scheme [10]	$\begin{array}{c} n(\left|H_{Z}\right|+\left|M_{G2}\right|+3\left|E_{G2}\right|+\\ s\left|M_{Z}\right|+s\left|A_{Z}\right|)\approx 2.4924n \end{array}$	$\begin{array}{c} cs\left|M_{Z}\right|+cs\left|A_{Z}\right|\\ \approx 0.009c \end{array}$	$\begin{array}{c} c\left(\left|M_{Z}\right|+\left|A_{Z}\right|\right)+s\left(\left|E_{G2}\right|+\left|M_{G2}\right|\right)\\ +2\left|P\right|\approx 0.0009c+2\left|P\right| \end{array}$
Scheme [11]	$\begin{array}{c} n(\left|H_{G2}\right|+3\left|E_{G2}\right|+\left|M_{G2}\right|)\\ \approx 3.61n \end{array}$	$\begin{array}{c} c(\left|M_{Z}\right|+\left|A_{Z}\right|+\\ \left|E_{G2}\right|+\left|M_{G2}\right|)\approx 0.8627c \end{array}$	$\begin{array}{c} c(\left|H\right|+2\left|M_{G2}\right|+2\left|E_{G2}\right|)+2\left|P\right|\\ \approx 1.7238c+2\left|P\right| \end{array}$
Scheme [12]	$\begin{array}{c} n(\left|H_{G2}\right|+\left|E_{G2}\right|\\ +\left|E_{Z}\right|)\approx 1.9601n \end{array}$	$\begin{array}{c} c(\left|M_{Z}\right|+\left|H_{G2}\right|+2\left|E_{G2}\right|+\\ 2\left|M_{G2}\right|+\left|A_{Z}\right|)\approx 2.0406c \end{array}$	2|P|
Scheme [13]	$\begin{array}{c} n(s+1)(\left|E_{G2}\right|+\left|M_{G2}\right|)\\ +n\left|H_{G2}\right|\approx 10.6066n \end{array}$	$\begin{array}{c} cs\left(\left|M_{Z}\right|+\left|A_{Z}\right|\right)+c\left|M_{G2}\right|\\ +c\left|E_{G2}\right|\approx 0.8708c \end{array}$	$\begin{array}{c} \left(c+s)(\left|M_{G2}\right|+\left|E_{G2}\right|\right)\\ +2\left|P\right|+c\left|H_{G2}\right|\approx 1.9886c+2\left|P\right| \end{array}$
Scheme [14]	$\begin{array}{c} n(2\left|A_{Z}\right|+\left|M_{Z}\right|)\\ \approx 0.0012n \end{array}$	$\begin{array}{c} c(2\left|M_{Z}\right|+\left|A_{Z}\right|+\left|E_{Z}\right|)\\ \approx 0.0241c \end{array}$	$\begin{array}{c} c\left|A_{Z}\right|+c\left|M_{Z}\right|+3\left|P\right|\\ \approx 0.0009c+3\left|P\right| \end{array}$
Our scheme	$\begin{array}{c} n(2\left|A_{Z}\right|+\left|M_{Z}\right|)\\ \approx 0.0012n \end{array}$	$\begin{array}{c} 3c\left|A_{Z}\right|\\ =0.0009c \end{array}$	$\begin{array}{c} c\left|A_{Z}\right|+c\left|A_{G1}\right|\\ \approx 0.0058c \end{array}$

## Data Availability

All relevant data has been provided in the article. If someone have any other needs, he or she can contact the authors by email.

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
