# Peer review of "SM2-Based Offline/Online Efficient Data Integrity Verification Scheme for Multiple Application Scenarios"

_sensors, 2023, doi:10.3390/s23094307_

Round 1
Reviewer 1 Report
The authors introduce a way to ensure that data stored on cloud storage are kept safe and secure. Safety verification pertains to another service that performs the check without having access to unencrypted data. The authors claim that their results apply to the remote storage of medical data.
Reading the paper is a demanding task. The main problem is in the content organization. Following are some of the issues I have found in the manuscript:
- the "preliminaries" section contains background material but also a high-level description of the architecture ("System Model"). Authors should separate the two topics. One solution envisions two different sections, not necessarily in that order. Figure 2 is very descriptive and can be anticipated, maybe before introducing the background material
- the algorithms listed after line 243 should be somewhat organized, and their introduction justified. For instance, "OnTagGen" is declared on page 6, defined on page 8, and used only in the proof of the third theorem. The reader should be able to connect this function with one of the arrows in figure2
- in my opinion, the game illustrated after line 268 plays a fundamental role, but it is poorly explained and, after being defined, it is used only once. Authors should better explain their point
- please explain or maybe fix the "Private Key Query" definition (line 273): consider that delivering a private key is contradictory, so you should elaborate on this
- overall readability would improve if the main content focuses on the theorem's relevance, while their proofs are moved to a dedicated appendix
Summarizing the paper is well-motivated and results may be relevant, but the authors should significantly improve its organization. The target is an article that conveys sufficient information even to a superficial reader and is exhaustive for a researcher that wants to use or improve the results.
Sometimes English is not fluent, but always understandable. The authors might consider using an automatic grammar checker. The typography of mathematics needs a fix.
Author Response
Dear reviewer, Thank you very much for your advice. We have revised the manuscript and would like to re-submit it for your consideration. We have addressed the comments raised by you, and the amendments are highlighted in red in the revised manuscript. Point-by-point responses to the comments are listed in this letter.

Reviewer 2 Report
I have had some problems following your proposal especially in Section 3 where many parameters are introduced without (IMHO) a proper discussion. Table I where parameters are just introduced should be placed at the beginning of Section 3.
Again in Section 3, I'm wondering why some math equations or symbols are in "superscript" format.
Concerning Section 6, I would suggest better explaining the results obtained and why they look like that. For example, why "the time cost of each
scheme increases as the number of data blocks increases, but the time cost of scheme [14] and our scheme does not increase significantly as the number of data blocks increases"? This section needs more discussion.
Author Response
Dear reviewer,
Thank you very much for your advice. We have revised the manuscript and would like to re-submit it for your consideration. We have addressed the comments raised by you, and the amendments are highlighted in red in the revised manuscript. Point-by-point responses to the comments are listed in the letter.

Reviewer 3 Report
Manuscript ID: sensors-2267363
Title: SM2-Based Offline/Online Efficient Data Integrity Verification Scheme for Multiple Application Scenarios
I admire the authors’ efforts for the preparation of this work and thank them for submitting this paper to the Sensors journal (ISSN:1424-8220). The article is fully eligible for publication in the Sensors journal.
Yours sincerely,
Author Response
Dear reviewer,
Thank you very much for your feedback. We appreciate your recognition of our research work. We have made some structural adjustments and grammatical optimizations to the manuscript to further enhance its readability.
Thank you again for your valuable comments, which have helped us to do a better job in our research.
Yours sincerely,
Xiuguang Li & Zhengge Yi.

Round 2
Reviewer 1 Report
The paper organization is satisfactory, but the English and typography should be further revised